# A Real-World Comparison of the Safety Profile for Immune Checkpoint Inhibitors in Oncology Patients

**DOI:** 10.3390/jcm14020388

**Published:** 2025-01-09

**Authors:** Abdulrahman Alwhaibi, Miteb A. Alenazi, Sultan Alghadeer, Wael Mansy, Reem A. Alsaif, Nawaf E. Abualreesh, Rakan J. Alanazi, Abdullah Alroumi, Saleh A. Alanazi

**Affiliations:** 1Department of Clinical Pharmacy, King Saud University, Riyadh 11451, Saudi Arabia; salghadeer@ksu.edu.sa (S.A.); wsayed@ksu.edu.sa (W.M.); na.abualreesh@gmail.com (N.E.A.); 2Pharmacy Department, King Saud University Medical City, Riyadh 11411, Saudi Arabia; mitalanazi@ksu.edu.sa; 3Pharmacy Department, King Abdulaziz Medical City, National Guard Health Affairs, Riyadh 11426, Saudi Arabia; reemaasalsaif@gmail.com; 4Pharmacy Practice Department, College of Pharmacy, Alfaisal University, Riyadh 11533, Saudi Arabia; rjalanazi@alfaisal.edu; 5College of Pharmacy, King Saud Bin Abdulaziz University for Health Sciences, Riyadh 11461, Saudi Arabia; alroumi@ngha.med.sa (A.A.); anazis@ngha.med.sa (S.A.A.)

**Keywords:** atezolizumab, pembrolizumab, nivolumab, durvalumab, complications, adverse effects, side effects, onset

## Abstract

**Background/Objectives:** Owing to the growing use of immune checkpoint inhibitors (ICIs) in the treatment of cancer, a wide spectrum of toxicity has arisen among cancer patients. Yet, limited ICI toxicity-related research is currently conducted in our region. **Methods:** This is a retrospective observational study conducted on adult cancer patients who received at least one cycle of ICI single therapy. Toxicity profiles of different ICI monotherapies were described and compared, and their association with different risk factors was assessed. SPSS version 28 was used for statistical analyses, and *p* < 0.05 was considered statistically significant. **Results:** A total of 428 patients were treated with anti-PD1 (nivolumab [*n* = 221, 51.6%] and pembrolizumab [*n* = 126, 29.5%]) or anti-PD-L1 (atezolizumab [*n* = 78, 18.2%] and durvalumab [*n* = 3, 0.7%]). Pneumonia was the most common complication (10%), followed by acute kidney injury (AKI; 8.2%) and hepatitis (7.9%). The proportion of hepatitis cases was significantly higher among atezolizumab compared to nivolumab-, pembrolizumab-, and durvalumab-treated patients (17.95% vs. 7.7% vs. 2.4% vs. 0.0%, respectively; *p* < 0.001). Gastrointestinal complication (colitis) was detected in 3.3% of patients with a significant difference between treatment groups (4.5%, 1.6%, 1.3%, and 33.3% in nivolumab, pembrolizumab, atezolizumab, and durvalumab, respectively; *p* = 0.008). Cardiac complications occurred in 1.2% of patients with a significant difference between treatment groups (0.5% in the nivolumab, 3.8% in the atezolizumab, 33.3% in the durvalumab, and none in the pembrolizumab groups (*p* < 0.001)). Musculoskeletal side effects, including both arthralgia and fatigue, were the most-reported side effects by 39.5% of patients, with significantly higher arthralgia complainers only in nivolumab (7.7%) compared to other treatment groups (0%, 2.6%, and 0% in pembrolizumab, atezolizumab, and durvalumab, respectively, *p* = 0.007). Hepatic, cardiovascular, hematological, respiratory, renal, gastrointestinal complications, thyroid complications, and dermatological side effects were found to occur on weeks 6, 7.5, 8, 8, 10, 10, 10.5, and 12 after treatment initiation, respectively, with no significant difference between treatment groups. Despite that, hepatitis and AKI tended to occur earlier with atezolizumab (week 2, *p* = 0.084) and pembrolizumab (week 2, *p* = 0.062), respectively, compared to their comparators. The female gender and a history of hepatitis were found to increase the odds of hepatic complication with anti-PD1 or anti-PD-L1 use [OR = 2.71; 95% CI 1.07–6.85, OR = 11.14; 95% CI 3.46–35.88, respectively]. Previous exposure to cancer therapy only was found to increase the odds of developing pneumonia among the treated patients [OR = 3.08; 95% CI 1.12–8.85]. Having hematological malignancy influenced the odds of hematological complications positively (either neutropenia or thrombocytopenia) compared to solid malignancies when patients were treated with anti-PD1 or anti-PD-L1 [OR = 17.18; 95% CI 4.06–72.71]. Finally, the female gender was found to positively associate with the odds of nausea/vomiting and fatigue secondary to anti-PD1 or anti-PD-L1 administration [OR = 2.08; 95% CI 1.34–3.21, OR = 1.65; 95% CI 1.09–2.51, respectively]. On the other hand, previous exposure to cancer therapy was found to reduce the risk of having arthralgia with anti-PD1 or anti-PD-L1 administration [OR = 0.344; 95% CI 0.121–0.974]. **Conclusions:** Treatment with anti-PD1 or anti-PD-L1 was associated with a spectrum of complications and side effects. Several risk factors have been identified to impact their occurrence. ICI toxicities and risk factors influencing their odds should be recognized and considered in clinical practice, as this could help in individualizing therapeutics regimens and avoiding treatment interruption.

## 1. Introduction

Recent advances in the field of immunotherapy have led to the development of immune checkpoint inhibitors (ICIs) [1], a group of medications that reshaped the landscape of cancer therapy with unprecedented success in the treatment of different types of cancer [2]. Their novel mechanism is rooted in targeting small proteins, cytotoxic T-lymphocyte antigen 4 (CTLA-4)/programmed cell death protein 1 (PD-1), which are expressed on T cells, B cells, and innate immune cells and programmed cell death ligand-1 protein (PD-L1), which is expressed on cancer cells [3,4]. The activation of these inhibitory proteins is critical for regulating immune response, which cancer cells can exploit to evade immunosurveillance [3,4]. Despite the growing use of anti-CTLA-4 (ipilimumab), anti-PD-1 (nivolumab, pembrolizumab, cemiplimab), and anti-PD-L1 (atezolizumab, avelumab, durvalumab) [5], a vast spectrum of inflammatory responses has been reported with their use (i.e., immune-related adverse events (irAEs)) owing to energizing T cells and, consequently, harnessing the host immune system [6]. These irAEs impact numerous systems, including but not limited to neurological, endocrine, pulmonary, gastrointestinal, and cardiovascular systems [7].

Although safety data of ICIs are demonstrated in randomized controlled trials (RCTs), they are not well captured in the large-scale population, owing to the restricted criteria of the population included in these trials [8,9]. In other words, given the unique mechanism of ICIs, heterogeneous characteristics of patients (with respect to age, disease severity, and comorbidities), the administered doses, and the length of treatment with ICIs, the safety profile of ICIs is still evolving; thus, further studies on adverse effects are yet to be carried out [10]. Therefore, real-world data obtained via cohort studies and reports represent valuable sources of information in characterizing the toxicity profiles of such novel therapies [11,12].

Several observational studies have been published recently on adverse effects (complications and side effects) associated with the use of ICIs. A case series performed by Kao and colleagues investigated the neurological complications in 347 patients treated with pembrolizumab or nivolumab [13]. Although 2.9% were diagnosed with myopathy, neuropathy, retinopathy, or headache, non-neurological complications including hypothyroidism, colitis, and hepatitis were also found in a few patients; hence, corticosteroids, intravenous immunoglobulin, or plasma exchange were needed to treat these patients. A retrospective, pharmacovigilance study by Mikami and colleagues showed a broader spectrum of ICI-related neurological complications, including hypophysitis/hypopituitarism, encephalitis/myelitis, meningitis, Guillain–Barre syndrome, myasthenia gravis, vasculitis, and neuropathy [14]. Notably, the risk was higher with combination therapy that involved anti-CTLA4 compared to monotherapy of anti-PD1, anti-PD-L1, or anti-CTLA4. Hypothyroidism and thyrotoxicosis, followed by hypothyroidism, were revealed in Lee and colleagues’ retrospective study that was conducted on patients treated with pembrolizumab or nivolumab monotherapy and in combination with anti-CTLA4, emphasizing the importance of monitoring thyroid function while being treated with these therapies [15]. Another retrospective study published by Naidoo and colleagues showed an increase in the incidence of pneumonia among patients receiving anti-PD1/anti-PD-L1 monotherapy and/or in combination with anti-CTLA4 [16]. The occurrence of pneumonia was significantly higher in the combination compared to monotherapy, while no significant difference was found between anti-PD1 and anti-PD-L1 monotherapy. Diarrhea was also reported by Wang and colleagues to occur with ICIs in a retrospective study (single or combination therapy); however, its incidence was considered an independent predictor for improved survival of treated patients [17]. Fatigue and rash were also shown to occur with ICIs (anti-PD1, anti-CTLA4, and combination), as reported by HSU and colleagues [18]. Recently, thromboembolism, including venous and arterial thromboembolism, was investigated retrospectively by Ide and colleagues and confirmed in approximately 7% of patients (38/548) with ICIs (single and combination therapy with ICIs or other antineoplastic therapies); yet, it did not significantly impact the patients’ overall survival [19]. Other hematological toxicities such as thrombocytopenia, anemia, and pancytopenia were shown to occur with anti-PD1 and anti-PD-L1, as reported by Benkhadra and colleagues [20]. Furthermore, ICI-induced hepatitis and an increased risk of hepatitis B virus reactivation associated with ICI therapies were elucidated recently in retrospective studies conducted by Hountondji and colleagues and Yin and colleagues, respectively [21,22]. Furthermore, the elevation of cardiac enzymes and new incidences of cardiotoxicity (newly diagnosed cardiac disease) associated with single or combined ICIs were retrospectively reported by two studies conducted by She and colleagues and Waheed and colleagues, respectively [23,24]; thus, ICI-induced hepatitis and cardiotoxicity should be monitored in patients who are currently or previously treated with ICIs [25].

Despite the wealth of literature related to ICI complications among cancer patients treated in the West and the variety of studies providing such data with different levels of evidence (RCTs, observational, case series, and reports), the research on this area among our population is scanty, which is probably attributed to the increased cost and delay in adding these medications to hospital formularies [26]. Additionally, conducting studies in real-world practice and comparing complications associated with the use of different ICIs support the clinical evaluation of these medications and provide a valuable tool that certainly helps with estimating treatment costs and deciding whether to include ICIs in hospital formularies. Our research aims to assess and compare the complications and side effects associated with ICI use, particularly nivolumab, pembrolizumab, atezolizumab, and durvalumab monotherapies and evaluate their occurrence with risk factors among cancer patients who received these therapies.

## 2. Methods

After receiving approval from the institutional review board committee at King Khalid University Hospital (KKUH) and National Guard Hospital (NGH) (IRB Project No. E-22-7262; NRC23R-152-02), a retrospective, chart-based review of all patients aged ≥ 18 years diagnosed with cancer and received at least one dose of ICI at any of the two hospitals at any time up to the end of year 2022 was initiated. All patients aged < 18 years despite the use of any ICI were excluded from this study. Additionally, all patients aged ≥ 18 years who received or were on multiple ICIs were excluded from this study.

Baseline demographic information including age, gender, body mass index (BMI), comorbidities, type and stage of malignancy, previous cancer therapies, and clinical parameters prior to ICI initiation were collected. ICI-based protocols, the date of their initiation, doses, and route of administration were documented for each patient. Complications and side effects [27] that occurred after ICI initiation were noted for each patient. The onset of a complication and side effects, defined as the time from ICI initiation to a censored observation and documentation of the incident at the follow up, was calculated for every patient, whenever possible. Furthermore, the association between complications, side effects, and different baseline and clinical variables was determined.

### Statistical Analysis

Normal distribution of data was investigated using the Shapiro–Wilk test and Kolmogorov–Smirnov tests, and group comparison was conducted accordingly via non-parametric statistics (Kruskal–Wallis tests) or parametric statistics (one-way ANOVA) for continuous variables. Categorial data were compared and presented as frequencies and percentages, and any association between them and the subjects’ groups was assessed using the chi-squared test or Fisher’s exact test. Multivariate logistic regression was used to examine predictors of complications and side effects, and the odds ratio (OR) with a 95% confidence interval (95% CI) was used to describe this association. All statistical analyses were performed using SPSS software version 28 (IBM Corp., Armonk, NY, USA), and *p* < 0.05 was taken to indicate statistical significance.

## 3. Results

### 3.1. Patient Selection and Baseline Characteristics

Figure 1 shows a flowchart demonstrating the selection process of patients included in this study. A total of 465 patients were identified to receive ICI therapies at KKUH or NGH. Anti-PD1 (nivolumab and pembrolizumab) and anti-PD-L1 (atezolizumab and durvalumab) were the only ICIs found to be administered in both centers. Since we intended to compare single ICIs only, patients who received multiple ICIs, from different or the same pharmacological group, were excluded from this study (*n* = 37), while those who started and continued on the same single ICI medication were eligible and included in the analysis (*n* = 428). Overall, nivolumab-treated patients accounted for 51.6% (*n* = 221), followed by 29.5% of pembrolizumab- (*n* = 126), 18.2% of atezolizumab- (*n* = 78), and 0.7% of durvalumab-treated patients (*n* = 3).

The baseline characteristics of the patients and the comparison between treatment groups are summarized in Table 1. The median age (IQR) of patients is 64 (20) years. Male patients represented 65%, and approximately half of the patients had chronic diseases that are risk factors for hypertension. Other comorbidities, including stroke, cerebrovascular accident, immobility, neuropathic pain, depression, hypothyroidism, hyperthyroidism, hyperparathyroidism, chronic obstructive pulmonary disease, asthma, respiratory infection, gastroesophageal reflux disease, gastric varices, brucellosis, hepatitis, liver cirrhosis, cholangitis, chronic kidney disease, benign prostatic hyperplasia, venous thromboembolism, osteoarthritis, and anemia, were diagnosed in 31.8% of the cohort. Additionally, for 61.4% of the patients aged ≥ 60 years, 95.3% were diagnosed with solid malignancies, with stage IV malignancies being the prominent stage (75.3%), and 91.8% and 91.2% had not suffered from cardiac disease or hepatitis, respectively, with a statistically significant difference in these variables detected between treatment groups. Approximately, 74% of the patients had previously received antineoplastic therapies other than ICIs, either targeted or chemotherapies, with no significant difference detected between treatment groups. The median body mass index (BMI), alanine transaminase (ALT), aspartate transaminase (AST), and bilirubin levels were 24.5 kg/m^2^, 20.5 U/L, 21 U/L, and Umol/L, with a statistically significant variation in their levels when compared among treatment groups. Further information about the baseline characteristics in each group is shown in Table 1.

### 3.2. Complications and Side Effects Associated with Anti-PD1 or Anti-PD-L1 Administration

Complications are undesirable, unpredicted adverse events that occur with the appropriate dose of medications and are lowered by reducing the dose or medication stoppage [27]. The rates of complications associated with anti-PD1 and anti-PD-L1 administration are depicted in Figure 2A,B. Of the complications, hepatitis was found in 7.9% of the cohort, with a significantly higher proportion among atezolizumab compared to nivolumab-, pembrolizumab-, and durvalumab-treated patients (17.95% vs. 7.7% vs. 2.4% vs. 0.0%, respectively; *p* < 0.001), as demonstrated in Table 2. Colitis, representing the gastrointestinal complication, was detected in 3.3% of the cohort, with a significant difference in its occurrence between treatment groups (4.5%, 1.6%, 1.3%, and 33.3% in nivolumab, pembrolizumab, atezolizumab, and durvalumab, respectively). Cardiovascular complications (atrial fibrillation, myocardial infarction (MI), cardiac arrest, and venous thromboembolism) occurred in 1.2% of patients, with a statistically significant difference in the proportion between treatment groups (0.5% in the nivolumab, 3.8% in the atezolizumab, 33.3% in the durvalumab, and none in the pembrolizumab groups (*p* < 0.001)). Respiratory (pneumonia), renal (AKI), hematological (neutropenia/thrombocytopenia), and thyroid complications (hypothyroidism/hyperthyroidism) were detected in 10.0%, 8.2%, 5.4%, and 4.2% of the whole patients, respectively, with no significant difference between the treatment groups. Other complications such as pleural effusion, osteoarthritis, cholecystitis, and anxiety were found in 3.7%, with a significant difference in its occurrence in atezolizumab compared to pembrolizumab, nivolumab, and durvalumab groups (9.0% vs. 4.0% vs. 1.8% vs. 0.0%, respectively; *p* = 0.039). Further details about the occurrence of complications overall and in each group are shown in Figure 2A,B and Table 2.

Unlike the complications, lowering medication is not required to resolve side effects, as they resolve on their own. Patients can adapt and get used to them, as they are usually counseled to probably experience them when medications are started. Side effects’ frequencies associated with the use of anti-PD1 or anti-PD-L1 are demonstrated in Figure 3A,B. Musculoskeletal side effects, including arthralgia and fatigue, were found to occur in 39.5% of the cohort, with a significant difference only detected in arthralgia between treatment groups (7.7%, 0%, 2.6%, and 0% in nivolumab, pembrolizumab, atezolizumab, and durvalumab, respectively, *p* = 0.007), as shown in Table 3. Gastrointestinal side effects presented by nausea, vomiting, and diarrhea only were detected in 35.7% of the whole group, with no significant difference between treatment groups. Dermatological side effects or skin reactions such as pruritis, itchiness, and eczema were reported by 7.5% of the patients, yet no significant difference was found between treatment groups. Further information related to the occurrence of side effects overall and in each group is shown in Figure 3A,B and Table 3.

Patients were classified according to the total number in Figure 3, which shows side effect occurrence in the study cohort according to the affected physiological system/organ.

Complications and side effects were then categorized and compared according to ICI therapies, as shown in Table 4. Overall, there is no significant difference between treatment groups in the frequency of those suffering from single or multiple complications or side effects. Further information is provided in Table 4.

### 3.3. Onset of Complications/Side Effects Associated with Anti-PD1 or Anti-PD-L1

The median number of weeks elapsed after treatment commencement until every complication or side effect occurrence was counted for each patient is depicted in Table 5. Hepatitis was found to occur the earliest (week 6), while dermatological effects as the only selected side effects occurred late (week 12) when compared to other complications, yet no significant difference was found in their onsets when inter-group comparison was conducted; *p* = 0.084 and *p* = 0.414, respectively. Despite that, hepatitis and AKI tended to occur earlier with atezolizumab (week 2, *p* = 0.084) and pembrolizumab (week 2, *p* = 0.062) when compared to other medications. Further details about the onset of each complication are provided in Table 5.

### 3.4. Assessment of Risk Factors Triggering Complications and Side Effect Occurrence with Anti-PD1 or Anti-PD-L1

Multivariate logistic regression analyses were carried out to assess the correlation between the baseline characteristics of the patients and each complication or side effect. With respect to hepatic complications (hepatitis), the female gender and a history of hepatitis were found to increase its odds upon treatment with anti-PD1 or anti-PD-L1 [OR = 2.71; 95% CI 1.07–6.85, OR = 11.14; 95% CI 3.46–35.88, respectively], whereas previous exposure to cancer therapy was the only factor that increased the odds of having respiratory complications (pneumonia) among the treated patients [OR = 3.08; 95% CI 1.12–8.85]. Diagnosis with hematological malignancy was found to significantly increase the odds of having hematological complications (either neutropenia or thrombocytopenia) when patients are treated with anti-PD1 or anti-PD-L1 compared to solid malignancies [OR = 17.18; 95% CI 4.06–72.71]. With respect to the side effects, the female gender was found to be positively associated with the odds of nausea/vomiting, musculoskeletal side effects, and fatigue, in particular, secondary to anti-PD1 or anti-PD-L1 administration [OR = 2.08; 95% CI 1.34–3.21, OR = 1.65; 95% CI 1.09–2.51, respectively]. On the other hand, previous exposure to cancer therapy was found to reduce the risk of having arthralgia with anti-PD1 or anti-PD-L1 administration [OR = 0.344; 95% CI 0.121–0.974]. None of the other baseline characteristics were found to significantly impact the odds of other complications and side effects when the same multivariate logistic regression analysis was carried out. Further details about the odds of having complications and side effects are provided in Table 6.

## 4. Discussion

Adverse effects, including complications and side effects, associated with the use of nivolumab, pembrolizumab, atezolizumab, and durvalumab for several types of cancer were investigated and compared in our study. Overall, approximately 70.8% (*n* = 303) of the whole cohort experienced adverse effects, of which 49.8% (*n* = 151), 16.8% (*n* = 51), and 33.3% (*n* = 101) experienced side effects only, complications only, and both, respectively. Respiratory (pneumonia), renal (AKI), and hepatic (hepatitis) events were ranked on the top list of complications at 10.0%, 8.2%, and 7.9%, respectively. With respect to side effects, musculoskeletal (fatigue) and gastrointestinal (nausea and vomiting) symptoms were found to be the most common among the study population, with values of 38.3% and 35.7%, respectively. In general, no significant difference in the proportion of those suffering from single or multiple side effects or complications was observed between treatment groups.

Despite the reported rates of complications secondary to the use of anti-PD1 or anti-PD-L1 in randomized clinical trials [28,29,30], variation in their rates was found later in real-world studies. For example, the incidence of pneumonia, hepatitis, AKI, hypothyroidism, and neurotoxicity (including neuropathies) was estimated to be 5%, 2–5%, 2%, 4–10%, and 6.1%, respectively based on randomized clinical trials [31,32]. Cardiac complications, such as myocarditis and heart failure, which were commonly seen with the combination of ICI blockade, and hematological toxicities, such as neutropenia and thrombocytopenia, were uncommon and observed in <1% and <10%, respectively [33,34]. Whereas, in a retrospective analysis of 139 case series and reports where nivolumab solely accounted for 77 the cases, the most common observed complications secondary to ICI usage were diabetes mellitus type-1 (15.8%; *n* = 22), AKI (11.5%; *n* = 16), colitis (10.1%; *n* = 14), hypophysitis (8.6%; *n* = 12), hepatitis (7.9%; *n* = 11), MI (7.2; *n* = 10), and hypothyroidism (5%; *n* = 7) [35], reiterating the notion that different rates of complications exist in the clinical trials compared to real-world studies. In our study, respiratory complication, i.e., pneumonia, was the most commonly observed among patients, accounting for 10% of total complications, followed by 8.2% renal (AKI), 7.9% hepatic (hepatitis), 5.4% hematological (neutropenia and thrombocytopenia), 4.2% thyroid (hypothyroidism and hyperthyroidism), 3.3% gastrointestinal (colitis), and 1.2% cardiovascular complications.

In a systematic review of 23 trials comparing the adverse events of anti-PD1 vs. anti-PD-L1 in 3284 patients treated for non-small-cell lung cancer, the incidence of complications including colitis, hypothyroidism, hepatitis, and nephritis tended to be higher in the anti-PD-1 group (16% vs. 11%; *p* = 0.07) [36]. Interestingly, although hypothyroidism was found the most common complication in both groups and presented with a higher rate in the anti-PD1 compared to the anti-PD-L1 group (6.7% vs. 4.2%, *p* = 0.7), the difference was not significant. On the other hand, a significant increase in respiratory complications, particularly pneumonitis, which was ranked as the second most commonly occurring complication, was seen in the anti-PD1 group (4% vs. 2%; *p* = 0.01); however, it is not clear whether this was driven by pembrolizumab, nivolumab, or both. Close to these findings, our study revealed that hypothyroidism occurs in 4.0% with a relatively similar rate in the anti-PD1 compared to the anti-PD-L1 group (4.0% vs. 3.7%), with no significant difference between nivolumab, pembrolizumab, atezolizumab and durvalumab monotherapies (3.2%, 5.6%, 3.8%, and 0%, *p* = 0.722, respectively). However, unlike being the most common complication reported previously [36], it was ranked the fourth complication in our study. Pneumonia was found to be the most common among our patients (10%). However, unlike the comparison in metanalysis, no significant difference was observed between anti-PD1 versus anti-PD-L1 groups (8.9% vs. 14.8%) or individual therapies (10.4%, 6.3%, 14.1%, and 33.3%, *p* = 0.161 in nivolumab, pembrolizumab, atezolizumab, and durvalumab, respectively). When logistic regression was conducted to predict the impact of patients’ characteristics on respiratory and thyroid complications, previous exposure to cancer therapy was discovered to increase the odds of pneumonia only [OR = 3.08; 95% CI 1.12–8.85], while none of the others happened to influence it. Additionally, none of the characteristics had a significant influence on the rate of hypothyroidism.

Unlike hypothyroidism and pneumonia, a significant increase in the rates of hepatitis (17.3% vs. 5.8%; *p* = 0.002) and cardiovascular complications (4.9% vs. 0.3%; *p* = 0.005) was observed in the anti-PD-L1 compared to the anti-PD1 group in our study. In detail, the proportion of those experiencing hepatitis after ICI initiation was significantly higher in atezolizumab compared to other groups (17.9% vs. 7.7% with nivolumab, 2.4% with pembrolizumab, 0% with durvalumab; *p* < 0.001). Interestingly, a previous history of hepatitis was more common among atezolizumab-treated patients compared to others. Upon assessing the predictors of hepatitis, the female gender and, more importantly, a previous history of hepatitis, were found to increase the odds of hepatitis [OR = 2.71; 95% CI 1.07–6.85, OR = 11.14; 95% CI 3.46–35.88, respectively]. Given the overall range of hepatitis, reported by a few studies to range between 6.1% and 14% [35,37,38], our finding falls in that range (7.9%). However, unlike the way we presented our results, these studies had no distinction in the rate of hepatitis between ICI medications due to either the absence of such comparison [35,38] or missing anti-PD-L1 agents [37]. Interestingly, Alnuhait and colleagues conducted a comparison of adverse events among ICI-treated patients in Saudi Arabia [39]. Although they showed a higher rate of hepatitis compared to ours (24.5% vs. 7.9%), the highest rate was found with atezolizumab (30%), followed by nivolumab (27%) and then pembrolizumab, which goes hand in hand with the trend seen in our patients. With respect to cardiovascular complications, they were found in several retrospective studies to range between 3.1% and 14.6% [9,24,35,38,40]. In our study, 1.2% of ICI-treated patients developed cardiovascular complications, and patients treated with durvalumab had the highest rate compared to others (33.3% vs. 3.8% with atezolizumab, 0.5% with nivolumab, 0% with pembrolizumab; *p* < 0.001), yet none of the baseline factors were found to impact their occurrence. Despite the low overall rate demonstrated here, the highest rate with durvalumab (33.3%) resonates with the study published by Waheed and colleagues, where durvalumab-treated patients had the highest rate compared to nivolumab and atezolizumab groups (25%, 15.2%, 8.9%, respectively) [24]. With respect to gastrointestinal complications, several reports have shown to occur in 0.8% to 15% [9,35,37,38,41,42], and the rate from our analysis (3.2%) falls in that range. Although the rate of colitis was significantly higher in durvalumab compared to others (33.3% vs. 4.5% with nivolumab, 1.6% with pembrolizumab, 1.3% with atezolizumab; *p* = 0.008), significance was lost when anti-PD-L1 was compared to the anti-PD1 group (2.5% vs. 3.5%, *p* > 0.05). Hematological complications including anemia, neutropenia, and thrombocytopenia were observed to reach 3.2% [38,39,42]. Our study showed that 5.4% of the patients have experienced neutropenia or thrombocytopenia, while none have had anemia. Another study showed that 19% of the patients developed thrombocytopenia/neutropenia only, while none were diagnosed with anemia post-ICI initiation [39]. In general, this variation in frequency is attributed to the widespread and evolving use of ICIs in multiple malignancies [43]. Upon analyzing the influence of baseline criteria on their incidence, hematological malignancies were predicted to increase the odds of hematological complications upon treatment with anti-PD1 and anti-PD-L1 [OR = 17.18; 95% CI 4.06–72.71].

With regard to side effects, our results seem to be consistent with those reported in clinical trials. In our study, three categories of side effects were reported from both anti-PD1 and anti-PD-L1 agents, namely, musculoskeletal (overall 39.5%; 38.3% for fatigue and 4.4% for arthralgia), gastrointestinal (overall 35.7%; 29.9% for nausea and vomiting and 12.4% for diarrhea), and dermatological (7.5% pruritis, itchiness, and eczema) side effects. Although fatigue was the most common side effect, no significant difference was found between anti-PD1 compared to anti-PD-L1 (37.8% vs. 40.0%, *p* = 0.704) or monotherapies. This resonates with the results published in a systematic review of 125 clinical trials that included 20,128 patients and assessed the occurrence of 75 ICI-related adverse effects [44]. In that review, fatigue was the most frequently reported all-grade adverse effect (18.26%), in spite of showing a lower rate compared to our finding, with no significant difference among various anti-PD1 and anti-PD-L1 agents. Interestingly, despite the absence of a significant difference in musculoskeletal symptoms in our study, including arthralgia, between anti-PD1 and anti-PD-L1 (4.9% vs. 2.5%, *p* = 0.548), the proportion of those suffering from arthralgia in nivolumab-treated patients was significantly higher compared to other groups (7.7% vs. 0%, 2.6% and 0% in pembrolizumab, atezolizumab, and durvalumab, respectively, *p* = 0.007). Furthermore, although the incidences of dermatological and gastrointestinal side effects, particularly diarrhea, in our study are numerically different compared to the ones reported in metanalysis (7.5% vs. 10.6% and 12.4% vs. 9.47%, respectively), pruritis/dermatitis and diarrhea were the most common ones (7.5% and 12.4%, respectively) after fatigue, nausea, and vomiting, which goes hand in hand with the results demonstrated in the meta-analysis (10.6% and 9.47%, respectively). Similar results were recently reported by El Majzoub and colleagues, in which diarrhea and dermatitis were among the top four most common ICI-related adverse events following fatigue [37]. Overall, when predictors of musculoskeletal and gastrointestinal side effects were assessed in our study, the female gender was found to increase the odds of fatigue and nausea and vomiting and [OR = 1.65; 95% CI 1.09–2.51, OR = 2.08; 95% CI 1.34–3.21, respectively], while previous exposure to cancer therapy was found to reduce the odds of arthralgia among patients [OR = 0.034; 95% CI 0.121–0.974].

Multiple reasons could explain such differences in the reported toxicities or complications between ICI classes and particular anti-PD1 and anti-PD-L1 agents. For example, there is a difference in the studies designed to test ICI-related adverse effects, in which some were single-armed trials and others were non-randomized studies. Others such as the absence of drug safety assessment as the primary objective in some studies, the need for dose escalation of particular agents while being used as first-line or second-line therapies, differences in the follow-up periods between studies, which potentially influence the actual onset of side effects, scarcity studies specifying the adverse effects caused by each agent, and differences in sample size between treatment groups have certainly confounded the actual incidence rate of complications and side effects associated with ICI agents [45]. More importantly, the difference in pharmacodynamic and pharmacokinetic properties between anti-PD1 and anti-PD-L1 agents played a crucial role in determining the selectivity, binding affinity, and duration of action of these therapies, which surely fostered varying adverse effects profiles of these therapies [46,47].

Immune checkpoint inhibitor-related toxicities were shown to occur with a median onset of 2–16 weeks after treatment initiation [7]. However, based on real-world studies, these toxicities vary, depending on the ICI agent, affected organs, types of malignancies, and patients’ characteristics [48]. In our study, we found hepatic complications to occur in the sixth week after treatment commencement, which falls within the reported onset range published in studies that included anti-PD1 and anti-PD-L1 therapies [34,49,50]. Interestingly, hepatitis associated with the use of pembrolizumab in our study did occur in week 10 compared to nivolumab (week 6), which aligns with the delayed onset of hepatitis reported previously, 19 weeks vs. 4 weeks, respectively [51]. Although cardiovascular and hematological complications are considered uncommon [10,51], our findings revealed that they occur in weeks 7.5 and 8, respectively, which is approximately close to the previously reported onset (week 6) [52]. Although pneumonia was reported to occur in months 3 and 5 [16,32], our results showed that it occurred in week 10, which is in agreement with phase III randomized clinical trials that evaluated the efficacy of nivolumab in melanoma patients [53]. AKI, colitis, and hypothyroidism are found to occur during the 10th week post-treatment initiation. Although this onset is in close proximity with previous studies [34,51,54], other evidence shows delayed onset of AKI, particularly for up to 11.6 months [32,48]. With respect to dermatological toxicity, a few studies have shown an early onset of skin reaction within 3–4 weeks of treatment initiation [49,50], while our study revealed its delayed onset, as it occurred in week 12, reiterating the potential impact of patients, the type of cancer, and treatment properties on the onset of ICI-related toxicities [48].

Several limitations exist in our study. First, there were no patients treated with anti-CTLA4 in the cohort, which could potentially influence our results given the high profile of toxicity for anti-CTLA4 therapies [50]. Second, only three patients were found to receive durvalumab, which led to an unequal distribution in the sample between treatment groups; hence, there was sample size bias and the absence of a normal distribution of data. To overcome that, non-parametric statistics for continuous variables using the Kruskal–Wallis test were considered upon conducting the analysis. Furthermore, although multivariate binary logistic regression analysis was conducted on all ICI-treated patients regardless of the used agents, the appropriateness of models was checked and confirmed, as shown by the non-significance Nagelkerke R Square values. To further investigate the impact of sample size bias, we have excluded durvalumab’s group and re-analyzed the data using the three other treatment groups only. Out of all of the results, only seven major changes were observed upon comparison to the four treatment groups, including two in the baseline criteria, two in the frequency of complications, two in the onset of complications, and one in the factors influencing the odds of complications (Appendix A). Third, the lack of grading for the severity of complications or side effects is another limitation that should be addressed in future studies. Additionally, we have considered the category of cancer only, such as solid or hematological malignancies, in our comparison rather than the actual type of cancer. Including such information would certainly add more value to the comparative analyses between these therapies. Despite these limitations, having a reasonable sample size in addition to conducting the study at two institutions increase the scientific value, clinical relevance, and applicability of our study.

In summary, given the varied frequency and onset of adverse effects associated with anti-PD1 and anti-PD-L1 therapies, healthcare providers should always work proactively and be vigilant for their signs and symptoms. For example, regular testing for hepatitis B virus (HBV) status in ICI-treated patients who lack antiviral prophylaxis can help in preventing HBV reactivation [22]. Additionally, monitoring of patients’ TSH, free T4, and transaminases, as examples, could potentially help in avoiding or ameliorating endocrine and hepatic toxicities, respectively [55].

## 5. Conclusions

Treatment with anti-PD1 or anti-PD-L1 was associated with a spectrum of complications and side effects, of which colitis, hepatitis, cardiovascular complications, and arthralgia were observed to be significantly different between nivolumab, pembrolizumab, atezolizumab, and durvalumab in the real-world experience. Their onset ranges between 6 to 12 weeks after treatment initiation. Although previous exposure to cancer therapies was associated with reduced odds of having arthralgia, an increase in the odds of developing hepatic, respiratory, hematological, and gastrointestinal complications and fatigue were associated with the female gender, an increased BMI, a history of hepatitis, hematological malignancy, and previous exposure to cancer therapies. Recognizing ICI toxicities and risk factor profiles influencing its occurrence is crucial in clinical practice, as this could help in selecting the most appropriate ICIs, individualizing therapeutic regimens, and avoiding treatment interruption. Cumulatively, the integration of our findings would impact clinical practice in the oncology unit and have added value to the quality of care for patients treated with ICIs.

## Figures and Tables

**Figure 1 jcm-14-00388-f001:**
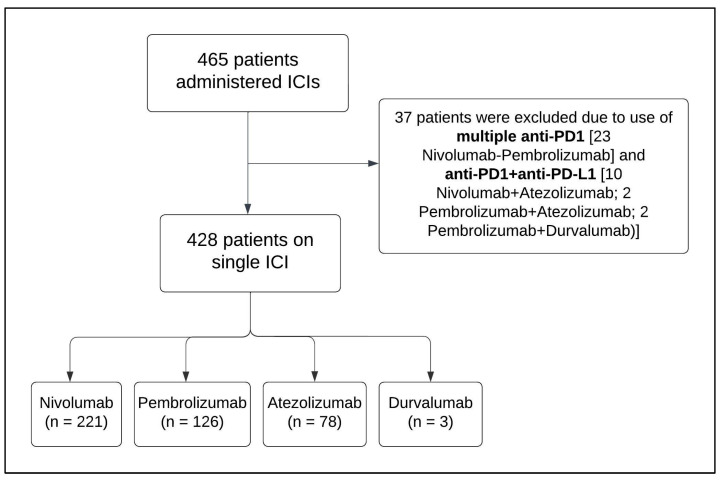
The flowchart demonstrates the selection process of patients included in this study (*n* = 428). Abbreviations: ICI: immune checkpoint inhibitor; PD1: programmed cell death protein 1; PD-L1: programmed cell death ligand-1 protein.

**Figure 2 jcm-14-00388-f002:**
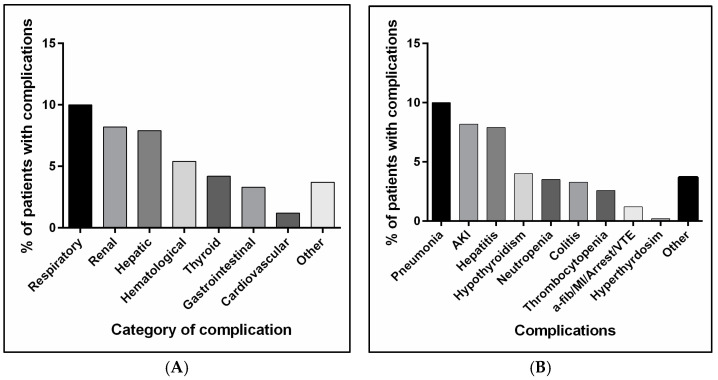
Complications occurrence in the study cohort. (**A**) represents the rate of complications classified based on the affected physiological system/organ. (**B**) represent the rate of each complication, except cardiovascular complications, which included one case of a-fib, MI, and cardiac arrest and two cases of VTE (one pulmonary embolism and one deep vein thrombosis). Abbreviations: AKI: acute kidney injury; VTE: venous thromboembolism.

**Figure 3 jcm-14-00388-f003:**
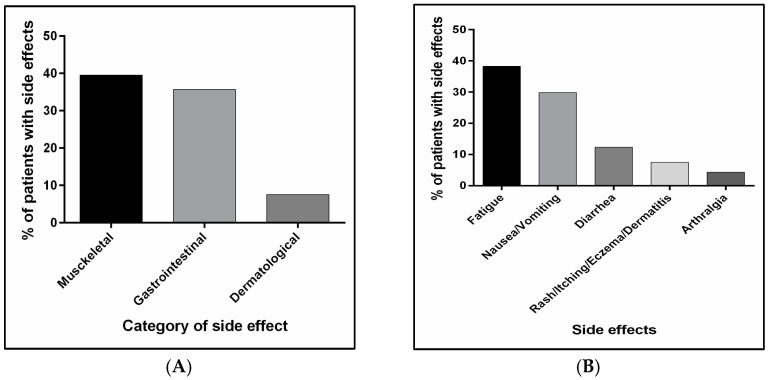
Side effect occurrence in the study cohort. (**A**) represents the rate of side effects classified based on the affected physiological system/organ. (**B**) represents the rate of each side effect, except nausea/vomiting and dermatological side effects.

**Table 1 jcm-14-00388-t001:** Baseline clinical characteristics of patients treated with ICIs.

Characteristics	All Patients (*n* = 428)	Nivolumab (*n* = 221)	Pembrolizumab (*n* = 126)	Atezolizumab (*n* = 78)	Durvalumab (*n* = 3)	*p*-Value
Age [median (IQR)]	64 (20)	64 (21)	61 (19)	64 (17.75)	67 (5.5)	0.147
Male gender, n (%)	278 (65.0)	154 (69.7)	73 (57.9)	48 (61.5)	3 (100)	0.075
Age group, n (%)						0.005
<30 yrs	11 (2.6)	10 (4.5)	1 (0.8)	0	0	
30 to <60 yrs	154 (36.0)	74 (33.5)	59 (46.8)	21 (26.9)	0	
≥60 yrs	263 (61.4)	137 (62.0)	66 (52.4)	57 (73.1)	3 (100.0)	
Primary diagnosed cancer, n (%) *						0.046
Solid	406 (95.3)	203 (92.7)	122 (96.8)	78 (100.0)	3 (100.0)	
Hematological	20 (4.7)	16 (7.3)	4 (3.2)	0	0	
Cancer stage right before ICI initiation, n (%) *						0.003
Stage II	40 (9.4)	20 (9.1)	8 (6.4)	12 (15.4)	0	
Stage III	65 (15.3)	47 (21.5)	9 (7.2)	9 (11.5)	0	
Stage IV	320 (75.3)	152 (69.4)	108 (86.4)	57 (73.1)	3 (100.0)	
Risk factors of heart diseases (HTN, DM, DLD, etc.)						0.335
Yes	232 (54.2)	115 (52.0)	72 (57.1)	42 (53.8)	3 (100.0)	
No	196 (45.8)	106 (48.0)	54 (42.9)	36 (46.2)	0	
Heart diseases						0.002
Yes	35 (8.2)	15 (6.8)	12 (9.5)	6 (7.7)	2 (66.7)	
No	393 (91.8)	206 (93.2)	114 (90.5)	72 (92.3)	1 (33.3)	
Heart diseases	35 (8.2)	15 (6.8)	12 (9.5)	6 (7.7)	2 (66.7)	
a-fib	2 (5.7)	1 (6.7)	1 (8.3)	0	0	0.882
IHD	21 (60)	7 (46.7)	8 (66.7)	4 (66.7)	2 (100.0)	0.427
CAD	3 (8.6)	2 (13.3)	1 (8.3)	0	0	0.757
CABG	4 (11.4)	2 (13.3)	1 (8.3)	1 (16.7)	0	0.899
CHF	5 (14.3)	3 (20.0)	1 (8.3)	1 (16.7)	0	0.775
Other comorbidities						0.091
Yes	136 (31.8)	75 (33.9)	31 (24.6)	30 (38.5)	0	
No	292 (68.2)	146 (66.1)	95 (75.4)	48 (61.5)	3 (100.0)	
History of hepatitis, n (%)						0.009
Yes	36 (8.4)	24 (10.9)	2 (1.6)	10 (12.8)	0	
No	392 (91.6)	197 (89.1)	124 (98.4)	68 (87.2)	3 (100.0)	
Previous cancer therapy, n (%)						0.565
Yes	316 (73.8)	165 (74.7)	94 (74.6)	54 (69.2)	3 (100.0)	
No	112 (26.2)	56 (25.3)	32 (25.4)	24 (30.8)	0	
BMI, kg/m^2^ [median (IQR)]	24.45 (7.8)	24.03 (7.9)	26.06 (9.6)	24.48 (5.8)	24.11 (0.24)	0.027
BMI, n (%)						0.014
Underweight	59 (13.8)	40 (18.1)	11 (8.7)	8 (10.3)	0	
Healthy weight	169 (39.5)	85 (38.5)	48 (38.1)	33 (42.3)	3 (100.0)	
Overweight	111 (25.9)	60 (27.1)	28 (22.2)	23 (29.5)	2 (40)	
Obesity	89 (20.8)	36 (16.3)	39 (31.0)	14 (17.9)	0	
Labs [median (IQR)]						
LDL (mmol/L) *	2.93 (1.0)	3.04 (1.56)	2.81 (1.27)	3.04 (1.08)	2.77 (0.49)	0.714
HDL (mmol/L) *	1.0 (0)	1.04 (0.36)	0.92 (0.31)	0.99 (0.23)	0.9 (0.26)	0.299
HbA1C% *	6.55 (3.0)	6.45 (3.13)	6.6 (2.7)	6.6 (2.5)	7.85 (0.85)	0.892
ALT (U/L) *	20.5 (20.8)	23.0 (20.75)	19.0 (16)	22.0 (24)	10 (3.5)	0.017
AST (U/L) *	21 (16.0)	22.0 (16.75)	18.0 (8)	23.0 (26)	16.0 (2)	<0.001
Bilirubin (Umol/L) *	8.9 (6.5)	9.5 (6.9)	7.77 (5.53)	10.0 (7.2)	7.1 (1.05)	0.003
Scr (Umol/L) *	70 (23)	70 (23)	69 (25)	71 (17)	74 (2)	0.857
CrCl (mL/min) *	78 (43.5)	74.5 (41.25)	85 (47)	78 (34)	70 (19.5)	0.148
Troponin (ng/mL)	5.4 (11)	4.65 (8.27)	9.55 (12.9)	6.73 (10.58)	7.85 (0)	0.341

Abbreviations: a-fib: atrial fibrillation; HbA1C%: hemoglobin A1C; ALT: alanine aminotransferase; AST: aspartate transaminase test; BMI: body mass index; CABG: coronary artery bypass grafting; CAD: coronary artery disease; CHF: congestive heart failure; CrCl: creatinine clearance; DM: diabetes mellitus; DLD: dyslipidemia; HDL: high-density lipoprotein; HTN: hypertension; ICI: immune checkpoint inhibitor; IQR: interquartile range; LDL: low-density lipoprotein; Scr: serum creatinine. * Missing data.

**Table 2 jcm-14-00388-t002:** Complications according to category and ICI medication.

Complication	All Patients (*n* = 428)	Nivolumab (*n* = 221)	Pembrolizumab (*n* = 126)	Atezolizumab (*n* = 78)	Durvalumab (*n* = 3)	*p*-Value
Respiratory, n (%)						0.161
Pneumonia						
Yes	43 (10.0)	23 (10.4)	8 (6.3)	11 (14.1)	1 (33.3)	
No	385 (90.0)	198 (89.6)	118 (93.7)	67 (85.9)	2 (66.7)	
Renal, n (%)						0.426
Acute kidney injury						
Yes	35 (8.2)	21 (9.5)	11 (8.7)	3 (3.8)	0	
No	393 (91.8)	200 (90.5)	115 (91.3)	75 (96.2)	3 (100.0)	
Hepatic, n (%)						<0.001
Hepatitis						
Yes	34 (7.9)	17 (7.7)	3 (2.4)	14 (17.9)	0	
No	394 (92.1)	204 (92.3)	123 (97.6)	64 (82.1)	3 (100.0)	
Hematological, n (%)						0.256
Yes	23 (5.4)	9 (4.1)	11 (8.7)	3 (3.8)	0	
No	405 (94.6)	212 (95.9)	115 (91.3)	75 (96.2)	3 (100.0)	
Hematological, n (%)						0.940
Neutropenia						
Yes	15 (3.5)	8 (3.6)	5 (4.0)	2 (2.6)	0	
No	413 (96.5)	213 (96.4)	121 (96.0)	76 (97.4)	3 (100.0)	
Thrombocytopenia						0.072
Yes	11 (2.6)	2 (0.9)	7 (5.6)	2 (2.6)	0	
No	417 (97.4)	219 (99.1)	119 (94.4)	76 (97.4)	3 (100.0)	
Thyroid, n (%)						0.692
Yes	18 (4.2)	7 (3.2)	7 (5.6)	4 (5.1)	0	
No	410 (95.8)	214 (96.8)	119 (94.4)	74 (94.4)	3 (100.0)	
Thyroid, n (%)						0.722
Hypothyroidism						
Yes	17 (4.0)	7 (3.2)	7 (5.6)	3 (3.8)	0	
No	411 (96)	214 (96.8)	119 (94.4)	75 (96.2)	3 (100.0)	
Hyperthyroidism						0.212
Yes	1 (0.2)	0	0	1 (1.3)	0	
No	427 (99.8)	221 (100.0)	126 (100.0)	77 (98.7)	3 (100.0)	
Gastrointestinal complications, n (%)						0.008
Colitis						
Yes	14 (3.3)	10 (4.5)	2 (1.6)	1 (1.3)	1 (33.3)	
No	414 (96.7)	211 (95.5)	124 (98.4)	77 (98.7)	2 (66.7)	
Cardiovascular, n (%)						<0.001
Yes	5 (1.2)	1 (0.5)	0	3 (3.8)	1 (33.3)	
No	423 (498.8)	220 (99.5)	126 (100.0)	75 (96.2)	2 (66.7)	
Others, n (%)						0.039
Yes	16 (3.7)	4 (1.8)	5 (4.0)	7 (9.0)	0	
No	412 (96.3)	217 (98.2)	121 (96.0)	71 (91.0)	3 (100.0)	

**Table 3 jcm-14-00388-t003:** Side effects according to category and ICI medication.

Side Effect	All Patients (*n* = 428)	Nivolumab (*n* = 221)	Pembrolizumab (*n* = 126)	Atezolizumab (*n* = 78)	Durvalumab (*n* = 3)	*p*-Value
Musculoskeletal, n (%)						0.663
Fatigue/arthralgia						
Yes	169 (39.5)	83 (37.6)	51 (40.5)	33 (42.3)	2 (66.7)	
No	259 (60.5)	138 (62.4)	75 (59.5)	45 (57.7)	1 (33.3)	
Musculoskeletal, n (%)						0.625
Fatigue						
Yes	164 (38.3)	80 (36.2)	51 (40.5)	31 (39.7)	2 (66.7)	
No	264 (61.7)	141 (63.8)	75 (59.5)	47 (60.3)	1 (33.3)	
Musculoskeletal, n (%)						0.007
Arthralgia						
Yes	19 (4.4)	17 (7.7)	0	2 (2.6)	0	
No	409 (95.6)	204 (92.3)	126 (100.0)	76 (97.4)	3 (100.0)	
Gastrointestinal, n (%)						0.144
Nausea/vomiting/diarrhea						
Yes	153 (35.7)	68 (30.8)	54 (42.9)	30 (38.5)	1 (33.3)	
No	275 (64.3)	153 (69.2)	72 (57.1)	48 (61.5)	3 (66.7)	
Gastrointestinal, n (%)						0.273
Nausea and vomiting						
Yes	128 (29.9)	59 (26.7)	42 (33.3)	27 (34.6)	0	
No	300 (70.1)	162 (73.3)	84 (66.7)	51 (65.4)	3 (100.0)	
Gastrointestinal, n (%)						0.324
Diarrhea						
Yes	53 (12.4)	22 (10.0)	19 (15.1)	11 (14.1)	1 (33.3)	
No	375 (87.6)	199 (90.0)	107 (84.9)	67 (85.9)	2 (66.7)	
Dermatological (pruritis–dermatitis–eczema), n (%)						0.298
Yes	32 (7.5)	18 (8.1)	9 (7.1)	4 (5.1)	1 (33.3)	
No	396 (92.5)	203 (91.1)	117 (92.9)	74 (94.9)	2 (66.7)	

**Table 4 jcm-14-00388-t004:** Comparison of total complications and side effects of ICI medications.

Characteristics	All Patients (*n* = 428)	Nivolumab (*n* = 221)	Pembrolizumab (*n* = 126)	Atezolizumab (*n* = 78)	Durvalumab (*n* = 3)	*p*-Value
Total complications						0.162
0	276 (64.5)	145 (65.6)	84 (66.7)	46 (59)	1 (33.3)	
1	122 (28.5)	62 (28.1)	38 (30.2)	21 (26.9)	1 (33.3)	
2	25 (5.8)	12 (5.4)	3 (2.4)	9 (11.5)	1 (33.3)	
3	4 (0.9)	2 (0.9)	1 (0.8)	1 (1.3)	0 (0.0)	
4	1 (0.2)	0 (0.0)	0 (0.0)	1 (1.3)	0 (0.0)	
Total side effects						0.173
0	176 (41.1)	100 (45.2)	47 (37.3)	29 (37.2)	0 (0.0)	
1	140 (32.7)	68 (30.8)	41 (32.5)	29 (37.2)	2 (66.7)	
2	82 (19.2)	33 (14.9)	34 (27.0)	14 (17.9)	1 (33.3)	
3	28 (6.5)	18 (8.1)	4 (3.2)	6 (7.7)	0 (0.0)	
4	2 (0.5)	2 (0.9)	0 (0.0)	0 (0.0)	0 (0.0)	

**Table 5 jcm-14-00388-t005:** Onset of complications and side effects according to ICI medication.

Complication/Side Effect	Average Week Elapsed Until Complication [Median (IQR)]	Weeks Elapsed Since Treatment Initiation Until Adverse Effect Occurrence [Median (IQR)]				*p*-Value
		Nivolumab (*n* = 221)	Pembrolizumab (*n* = 126)	Atezolizumab (*n* = 78)	Durvalumab (*n* = 3)	
Hepatic (hepatitis)	6.0 (14.0)	6.0 (12.5)	10.0 (31.5)	2.0 (3.5)	NA	0.084
Cardiovascular	7.5 (14.0)	NA	NA	5.0 (8.5)	10.0 (0.0)	0.655
Hematological	8.0 (26.0)	24.0 (31.0)	6.0 (9.5)	30.0 (6.5)	NA	0.145
Neutropenia	22.0 (30.0)	25.0 (35.25)	7.0 (18.0)	34.5 (4.5)	NA	0.174
Thrombocytopenia	7.0 (24.0)	4.0 (3.0)	6.0 (11.0)	32.5 (6.5)	NA	0.222
Respiratory (pneumonia)	8.0 (27.0)	10.0 (35.0)	11.0 (38.0)	4.0 (5.0)	8.0 (0.0)	0.182
Renal (AKI)	10.0 (15.0)	10.0 (9.0)	2.0 (3.0)	26.0 (10.5)	NA	0.062
Gastrointestinal complications (colitis)	10.0 (22.0)	8.5 (18.5)	6.0 (38.0)	16.0 (0.0)	27.0 (0.0)	0.631
Thyroid	10.5 (22.0)	8.0 (11.5)	23.0 (7.0)	6.0 (14.5)	NA	0.218
Hypothyroidism	12 (19)	8.0 (11.5)	23.0 (7.0)	9.0 (20)	NA	0.385
Skin reaction	12.0 (21.25)	11.5 (19.0)	10 (42)	18.5 (65)	NA	0.414

Missing data.

**Table 6 jcm-14-00388-t006:** Factors associated with complications and side effects of ICI medications.

Complication/Side Effect	Patients’ Factors		Coef (B)	S.E	Odds Ratio	95% CI	*p*-Value
Hepatic complication	Gender	Male			Ref		
		Female	0.996	0.473	2.71	1.07–6.85	0.035
	History of hepatitis	No			Ref		
		Yes	2.41	0.597	11.14	3.46–35.88	<0.001
Respiratory complication	Previous cancer therapy	No			Ref		
		Yes	1.125	0.515	3.08	1.12–8.85	0.029
Hematological complication	Type of malignancy	Solid			Ref		
		Liquid	2.84	0.736	17.18	4.06–72.71	<0.001
Neutropenia	Type of malignancy	Solid					
		Liquid	1.79	0.816	6.01	1.21–29.75	0.028
Thrombocytopenia	Type of malignancy	Solid					
		Liquid	3.28	1.08	26.5	3.22–218.38	0.002
Musculoskeletal	Gender	Male					
		Female	0.419	0.21	1.52	1.01–2.29	0.046
Arthralgia	Previous cancer therapy	No			Ref		
		Yes	−1.07	0.531	0.344	0.121–0.974	0.045
Fatigue	Gender	Male			Ref		
		Female	0.503	0.212	1.65	1.09–2.51	0.018
Nausea and vomiting	Gender	Male			Ref		
		Female	0.731	0.222	2.08	1.34–3.21	0.001

## Data Availability

Data presented in this study are available upon request from the corresponding author. Data are not publicly available due to privacy reasons.

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
