# Peer review of "A Real-World Comparison of the Safety Profile for Immune Checkpoint Inhibitors in Oncology Patients"

_jcm, 2025, doi:10.3390/jcm14020388_

Round 1

Reviewer 1 Report

Comments and Suggestions for Authors

The current study titled as “Real-world Comparison of Safety Profile for Immune-Check-2 point Inhibitors in Oncology Patients” is the study of relevance and interest for the journal readers. The authors have addressed the growing concern of immune checkpoint inhibitors (ICI)-related toxicities, particularly in a region with limited research data. As ICIs are increasing used in treatment of cancer worldwide.

I found the paper to be overall well written and felt confident that the authors carefully studied a significant research topic.

However, below are some of my comments that I want the authors to address prior to my endorsement for the publication of this manuscript ….

1. The sample size for durvalumab [n = 3, 0.7%] is quite small, can the author explain how it would not affect the generalizability of the finding?

2. The result section is abstract can be rewritten concisely without eliminating the important information. I mean it can be made less wordy.

3. Please number the headings and subheading for better understanding of the readers.

4. The caption of Figure 2 is not self-explanatory. Please mention what A and B are representing in the caption too, for better clarity.

5. The author has used the words adverse effects, complications and side effects. Would they explain or made distinction among them so that the readers are clear what conditions falls under these categories. For reference, please look first paragraph of the discussion section that has confused this reviewer too about these terms.

Author Response

Reviewer 1:

Comments and Suggestions for Authors

The current study titled as “Real-world Comparison of Safety Profile for Immune-Check-point Inhibitors in Oncology Patients” is the study of relevance and interest for the journal readers. The authors have addressed the growing concern of immune checkpoint inhibitors (ICI)-related toxicities, particularly in a region with limited research data. As ICIs are increasing used in treatment of cancer worldwide.

Comment 1: I found the paper to be overall well written and felt confident that the authors carefully studied a significant research topic.

Response 1: We really appreciate your comment.

Comment 2: The sample size for durvalumab [n = 3, 0.7%] is quite small, can the author explain how it would not affect the generalizability of the finding?

Response 2: Thank you for your comment. In fact, we have noticed an absence of durvalumab-treated patients in some studies (PMID: 33689534) and small number of durvalumab-treated in other studies (J Hematol Oncol Pharm. 2021;11(3):127-133). Because of that, researchers combined all anti-PDL1 medications together to avoid this issue. The same thing applies when small sample size present in any other agents belonging to anti-PD1 or anti-PDL1, hence they also combined even anti-PD1/anti-PDL1 (J Hematol Oncol Pharm. 2021;11(3):127-133) or all of ICIs together (PMID: 34327774). In our study, however, although this choice was doable, we preferred to keep them separated since we wanted to compare the safety profile of single therapies rather than groups and apply statistics that could overcome the small sample size of medications. Having the unequal sample size for treatment groups and absence of normal data distribution in our research, non-parametric statistics was the appropriate method and since we have >2 medication groups, Mann-Whitney U test could not be used and, rather, Kruskal Wallis tests would be the most appropriate test for continuous variables’ comparison, thus used in analyzing all continuous variables. Furthermore, we were concern if binary logistic regression analysis would be influenced; However, since all treated patients were considered as one group rather than dividing them in subgroups based on treatments, sample size would have no effect. Yet, regression model appropriateness was checked and confirmed using Nagelkerke R Square that showed no significance.

Comment 3: The result section in abstract can be rewritten concisely without eliminating the important information. I mean it can be made less wordy.

Response 3: We appreciate your suggestion. We have removed some phrases and tried to meake it less wordy as requested.

Comment 4: Please number the headings and subheading for better understanding of the readers.

Response 4: We appreciate your suggestion. Headings and subheading were numbered as requested.

Comment 5: The caption of Figure 2 is not self-explanatory. Please mention what A and B are representing in the caption too, for better clarity.

Response 5: We appreciate your comment. We have modified the caption in both Fig 2 and 3 for better clarity.

Comment 6: The author has used the words adverse effects, complications and side effects. Would they explain or made distinction among them so that the readers are clear what conditions falls under these categories. For reference, please look first paragraph of the discussion section that has confused this reviewer too about these terms.

Response 6: We apologize for this confusion. Complications and side effects both fall under adverse effects or adverse events. Adverse effects and adverse event are usually used interchangeably. Having said that, the word “complication” is used for the undesirable, unpredicted adverse events that occur with the appropriate dose of medications and are only lowered and resolved with reducing the dose or medication stoppage, otherwise they would harm the patients. Unlike the complications, lowering medication dose is not required for resolving side effects as they resolve with their own and patients can adapt and get used to them as they are counseled regarding the probability of experiencing them when medications are started. Please refer to reference [27], which we cited in methods and results sections. Hope this resolves the confusion.

Author Response:

The current study titled as “Real-world Comparison of Safety Profile for Immune-Check-2 point Inhibitors in Oncology Patients” is the study of relevance and interest for the journal readers. The authors have addressed the growing concern of immune checkpoint inhibitors (ICI)-related toxicities, particularly in a region with limited research data. As ICIs are increasing used in treatment of cancer worldwide.

I found the paper to be overall well written and felt confident that the authors carefully studied a significant research topic.

However, below are some of my comments that I want the authors to address prior to my endorsement for the publication of this manuscript ….

1. The sample size for durvalumab [n = 3, 0.7%] is quite small, can the author explain how it would not affect the generalizability of the finding?

2. The result section is abstract can be rewritten concisely without eliminating the important information. I mean it can be made less wordy.

3. Please number the headings and subheading for better understanding of the readers.

4. The caption of Figure 2 is not self-explanatory. Please mention what A and B are representing in the caption too, for better clarity.

5. The author has used the words adverse effects, complications and side effects. Would they explain or made distinction among them so that the readers are clear what conditions falls under these categories. For reference, please look first paragraph of the discussion section that has confused this reviewer too about these terms.

Author Response

Response to Reviewer 1 Comments

Reviewer 1:

Comments and Suggestions for Authors

The current study titled as “Real-world Comparison of Safety Profile for Immune-Check-point Inhibitors in Oncology Patients” is the study of relevance and interest for the journal readers. The authors have addressed the growing concern of immune checkpoint inhibitors (ICI)-related toxicities, particularly in a region with limited research data. As ICIs are increasing used in treatment of cancer worldwide.

Comment 1: I found the paper to be overall well written and felt confident that the authors carefully studied a significant research topic.

Response 1: We really appreciate your comment.

Comment 2: The sample size for durvalumab [n = 3, 0.7%] is quite small, can the author explain how it would not affect the generalizability of the finding?

Response 2: Thank you for your comment. In fact, we have noticed an absence of durvalumab-treated patients in some studies (PMID: 33689534) and small number of durvalumab-treated in other studies (J Hematol Oncol Pharm. 2021;11(3):127-133). Because of that, researchers combined all anti-PDL1 medications together to avoid this issue. The same thing applies when small sample size present in any other agents belonging to anti-PD1 or anti-PDL1, hence they also combined even anti-PD1/anti-PDL1 (J Hematol Oncol Pharm. 2021;11(3):127-133) or all of ICIs together (PMID: 34327774). In our study, however, although this choice was doable, we preferred to keep them separated since we wanted to compare the safety profile of single therapies rather than groups and apply statistics that could overcome the small sample size of medications. Having the unequal sample size for treatment groups and absence of normal data distribution in our research, non-parametric statistics was the appropriate method and since we have >2 medication groups, Mann-Whitney U test could not be used and, rather, Kruskal Wallis tests would be the most appropriate test for continuous variables’ comparison, thus used in analyzing all continuous variables. Furthermore, we were concern if binary logistic regression analysis would be influenced; However, since all treated patients were considered as one group rather than dividing them in subgroups based on treatments, sample size would have no effect. Yet, regression model appropriateness was checked and confirmed using Nagelkerke R Square that showed no significance.

Comment 3: The result section in abstract can be rewritten concisely without eliminating the important information. I mean it can be made less wordy.

Response 3: We appreciate your suggestion. We have removed some phrases and tried to meake it less wordy as requested.

Comment 4: Please number the headings and subheading for better understanding of the readers.

Response 4: We appreciate your suggestion. Headings and subheading were numbered as requested.

Comment 5: The caption of Figure 2 is not self-explanatory. Please mention what A and B are representing in the caption too, for better clarity.

Response 5: We appreciate your comment. We have modified the caption in both Fig 2 and 3 for better clarity.

Comment 6: The author has used the words adverse effects, complications and side effects. Would they explain or made distinction among them so that the readers are clear what conditions falls under these categories. For reference, please look first paragraph of the discussion section that has confused this reviewer too about these terms.

Response 6: We apologize for this confusion. Complications and side effects both fall under adverse effects or adverse events. Adverse effects and adverse event are usually used interchangeably. Having said that, the word “complication” is used for the undesirable, unpredicted adverse events that occur with the appropriate dose of medications and are only lowered and resolved with reducing the dose or medication stoppage, otherwise they would harm the patients. Unlike the complications, lowering medication dose is not required for resolving side effects as they resolve with their own and patients can adapt and get used to them as they are counseled regarding the probability of experiencing them when medications are started. Please refer to reference [27], which we cited in methods and results sections. Hope this resolves the confusion.

Reviewer 2 Report

Comments and Suggestions for Authors

The manuscript shows an interesting retrospective for an observational study comparing the safety profiles of various immune checkpoint inhibitors in oncology patients. The data collection, statistical analysis, and presentation seem accurate but require some improvements for clarity.  

The design of the study is suitable for assessing real-world complications and side effects, but I  recommend more elaborated exclusion criteria and how confounding variables were controlled. Some of the limitations section acknowledges the lack of anti-CTLA4 data and small sample size for durvalumab, which weakens comparisons. Including broader cohorts would strengthen conclusions. Address why complications, such as cardiovascular events, varied widely among small cohorts like durvalumab (n=3).

Data were analyzed using appropriate statistical tests, but the additional justification for the chosen methods like the logistic regression would improve transparency.

The lack of grading for the severity of complications is a limitation that should be addressed in future studies to provide more insights. Describe how statistical significance was determined discuss potential biases due to sample size disparities, and include sensitivity analyses to validate findings.

Author Response

Reviewer 2:

Comments and Suggestions for Authors

The manuscript shows an interesting retrospective for an observational study comparing the safety profiles of various immune checkpoint inhibitors in oncology patients. The data collection, statistical analysis, and presentation seem accurate but require some improvements for clarity. The design of the study is suitable for assessing real-world complications and side effects, but

Comment 1: I recommend more elaborated exclusion criteria

Response 1: We appreciate your comment. We have modified part of the method as the following: a retrospective, chart-based review of all patients aged ≥ 18 years, diagnosed with cancer and received at least one dose of ICIs at any of the two hospitals at any time up to the end of year 2022 was initiated. All patients aged <18 years despite the use of any ICI were excluded from the study. Additionally, all patients aged ≥ 18 years and receiving or being on multiple ICIs were excluded form the study.

Hope this made it clearer for the readers.

Comment2: I recommend more elaboration on how confounding variables were controlled.

Response 2: We appreciate your comment. Sample size and lack of data distribution were among the major drawbacks of our study. Having the unequal sample size for treatment groups and absence of normal data distribution in our research, non-parametric statistics was the appropriate method and since we have >2 medication groups, Mann-Whitney U test could not be used and, rather, Kruskal Wallis tests would be the most appropriate test for continuous variables’ comparison, thus used in analyzing all continuous variables. Furthermore, we were concern if binary logistic regression analysis would be influenced; However, since all treated patients were considered as one group rather than dividing them in subgroups based on treatments, sample size would have no effect. Yet, regression model appropriateness was checked and confirmed using Nagelkerke R Square that showed no significance. This has been added to the limitation of study.

Comment 3: Some of the limitations section acknowledges the lack of anti-CTLA4 data and small sample size for durvalumab, which weakens comparisons. Including broader cohorts would strengthen conclusions.

Response 3: We appreciate your comment. We have addressed that in the limitations of the study.

Comment 4: Address why complications, such as cardiovascular events, varied widely among small cohorts like durvalumab (n=3).

Response 4: We appreciate your comment. This is attributed to the low sample size of durvalumab. Please note that we have noticed an absence of durvalumab-treated patients in some studies (PMID: 33689534) and small number of durvalumab-treated in other studies (J Hematol Oncol Pharm. 2021;11(3):127-133). Because of that, researchers combined all anti-PDL1 medications together to avoid this issue. The same thing applies when small sample size present in any other agents belonging to anti-PD1 or anti-PDL1, hence they also combined even anti-PD1/anti-PDL1 (J Hematol Oncol Pharm. 2021;11(3):127-133) or all of ICIs together (PMID: 34327774). In our study, however, although this choice was doable, we preferred to keep them separated since we wanted to compare the safety profile of single therapies rather than groups and apply statistics that could overcome the small sample size of medications.

Comment 5: Data were analyzed using appropriate statistical tests, but the additional justification for the chosen methods like the logistic regression would improve transparency.

Response 5: We appreciate your comment. We have elaborated on that in Line 452-460.

Comment 6: The lack of grading for the severity of complications is a limitation that should be addressed in future studies to provide more insights.  

Response 6: We appreciate your comment. We have incorporated that in the limitation section.

Comment 7: Describe how statistical significance was determined; Discuss potential biases due to sample size disparities; and include sensitivity analyses to validate findings.

Response 6: We appreciate your comment. Statistical significance was determined if p value was < 0.05 for continuous and categorial analyses. This was written in the method section. Please refer to the method section for further explanation (Line 150-159). The effect of sample size and how to consider that in analysis was taken into consideration and addressed in the limitation. Please refer to Line 452-460.

Author Response

Response to Reviewer 2 Comments

The manuscript shows an interesting retrospective for an observational study comparing the safety profiles of various immune checkpoint inhibitors in oncology patients. The data collection, statistical analysis, and presentation seem accurate but require some improvements for clarity. The design of the study is suitable for assessing real-world complications and side effects, but

Comment 1: I recommend more elaborated exclusion criteria

Response 1: We appreciate your comment. We have modified part of the method as the following: a retrospective, chart-based review of all patients aged ≥ 18 years, diagnosed with cancer and received at least one dose of ICIs at any of the two hospitals at any time up to the end of year 2022 was initiated. All patients aged <18 years despite the use of any ICI were excluded from the study. Additionally, all patients aged ≥ 18 years and receiving or being on multiple ICIs were excluded form the study.

Hope this made it clearer for the readers.

Comment2: I recommend more elaboration on how confounding variables were controlled.

Response 2: We appreciate your comment. Sample size and lack of data distribution were among the major drawbacks of our study. Having the unequal sample size for treatment groups and absence of normal data distribution in our research, non-parametric statistics was the appropriate method and since we have >2 medication groups, Mann-Whitney U test could not be used and, rather, Kruskal Wallis tests would be the most appropriate test for continuous variables’ comparison, thus used in analyzing all continuous variables. Furthermore, we were concern if binary logistic regression analysis would be influenced; However, since all treated patients were considered as one group rather than dividing them in subgroups based on treatments, sample size would have no effect. Yet, regression model appropriateness was checked and confirmed using Nagelkerke R Square that showed no significance. This has been added to the limitation of study.

Comment 3: Some of the limitations section acknowledges the lack of anti-CTLA4 data and small sample size for durvalumab, which weakens comparisons. Including broader cohorts would strengthen conclusions.

Response 3: We appreciate your comment. We have addressed that in the limitations of the study.

Comment 4: Address why complications, such as cardiovascular events, varied widely among small cohorts like durvalumab (n=3).

Response 4: We appreciate your comment. This is attributed to the low sample size of durvalumab. Please note that we have noticed an absence of durvalumab-treated patients in some studies (PMID: 33689534) and small number of durvalumab-treated in other studies (J Hematol Oncol Pharm. 2021;11(3):127-133). Because of that, researchers combined all anti-PDL1 medications together to avoid this issue. The same thing applies when small sample size present in any other agents belonging to anti-PD1 or anti-PDL1, hence they also combined even anti-PD1/anti-PDL1 (J Hematol Oncol Pharm. 2021;11(3):127-133) or all of ICIs together (PMID: 34327774). In our study, however, although this choice was doable, we preferred to keep them separated since we wanted to compare the safety profile of single therapies rather than groups and apply statistics that could overcome the small sample size of medications.

Comment 5: Data were analyzed using appropriate statistical tests, but the additional justification for the chosen methods like the logistic regression would improve transparency.

Response 5: We appreciate your comment. We have elaborated on that in Line 452-460.

Comment 6: The lack of grading for the severity of complications is a limitation that should be addressed in future studies to provide more insights.  

Response 6: We appreciate your comment. We have incorporated that in the limitation section.

Comment 7: Describe how statistical significance was determined; Discuss potential biases due to sample size disparities; and include sensitivity analyses to validate findings.

Response 6: We appreciate your comment. Statistical significance was determined if p value was < 0.05 for continuous and categorial analyses. This was written in the method section. Please refer to the method section for further explanation (Line 150-159). The effect of sample size and how to consider that in analysis was taken into consideration and addressed in the limitation. Please refer to Line 452-460.

Round 2

Reviewer 2 Report

Comments and Suggestions for Authors

The new adjustments are clear, and several comments from the initial review were addressed, but there are still a few things that need improvement, such as the grading severity, while it is mentioned as a limitation, the absence of severity grading significantly reduces the clinical applicability of the findings, maybe use a supplementary analysis using available data might mitigate this gap.

Authors noted some biases in their sample size and statistical tests, but sensitivity analyses were not explicitly included, by conducting those tests could reinforce the findings in the manuscript.

The discussion could expand on implications for patient monitoring and decision-making in real-world oncology settings. I mean,  expanding more about how these findings can guide clinicians in monitoring ICIs is underdeveloped.

Author Response

Comments and Suggestions for Authors

Comment 1: The grading severity, while it is mentioned as a limitation, the absence of severity grading significantly reduces the clinical applicability of the findings, maybe use a supplementary analysis using available data might mitigate this gap.

Response 1: Thank you for your comment. We absolutely agree on how absence the grading of adverse effects reduces the clinical applicability of the findings of our study. Unfortunately, we have missed adding adverse effect grading when data collection sheet was designed thus was not collected during data collection step. Additionally, having two cancer centers used in the study, there was inconsistency in grading severity. We have gone through some patients charts after receiving this comment but we could not find it which hindered our ability to consider it in the analysis.

Comment 2: Authors noted some biases in their sample size and statistical tests, but sensitivity analyses were not explicitly included, by conducting those tests could reinforce the findings in the manuscript.

Response 2: Thank you for your comment. We acknowledge the sample size bias, yet this is expected given the retrospective nature of the study. One of the ways to overcome this is to increase the size of groups that have low number of patients. Although this sounds doable, it will be challenging given the time required for IRB approval and data collection. Since durvalumab’s groups has the lowest number of patients and any change in complications or side effects frequency will appear larger than other groups, we have excluded this group and re-analyzed the data. When we compared the results based on 3 treatment groups to 4 treatment groups, only 7 major changes were observed as listed below. This was described in the manuscript (discussion; lines 459-464) and tables were added as supplementary 1.

Table 1. Baseline clinical characteristics of patients treated with ICIs.

Characteristics

Nivolumab (n=221)

Pembrolizumab (n=126)

Atezolizumab (n=78)

P-value

Heart diseases

Yes

No

15 (6.8)

206 (93.2)

12 (9.5)

114 (90.5)

6 (7.7)

72 (92.3)

0.657

Labs [Median (IQR)]

ALT (U/L)*

24.03 (7.9)

26.06 (9.6)

24.48 (5.8)

0.06

Table 2. Complications according to category and ICI medication.

Complication

Nivolumab (n=221)

Pembrolizumab (n=126)

Atezolizumab (n=78)

P-value

Thrombocytopenia

Yes

No

2 (0.9)

219 (99.1)

7 (5.6)

119 (94.4)

2 (2.6)

72 (97.4)

0.032

Gastrointestinal complication (colitis)

Yes

No

10 (4.5)

211 (95.5)

2 (1.6)

124 (98.4)

1 (1.3)

77 (98.7)

0.187

Table 5. Onset of complications and side effects according to ICI medication

Complication/side effect

Average Weeks elapsed until complication

[Median (IQR)]

Weeks elapsed since treatment initiation until adverse effects occurrence

[Median (IQR)]

Nivolumab (n=221)

Pembrolizumab (n=126)

Atezolizumab (n=78)

P-value

Cardiovascular

5.0 (NA)

NA

NA

5.0 (8.5)

NA

Gastrointestinal complications (colitis)

8.5 (19)

8.5 (18.5)

6.0 (38.0)

16.0 (0.0)

0.368

Table 6. Factors associated with complications and side effects of ICI medications.

Complication/side effect

Patients’ factors

Coef (B)

S.E

Odds ratio

95% CI

P value

Hepatic complication

Gender

Male

Ref

Female

0.797

0.455

2.21

0.91 – 5.41

0.08

Comment 3: The discussion could expand on implications for patient monitoring and decision-making in real-world oncology settings. I mean, expanding more about how these findings can guide clinicians in monitoring ICIs is underdeveloped.

Response 3: Thank you for your comment. We have elaborated on that in the discussion as requested. Please refer to Line 472-478.
